# The public health sector supply chain costs in Tanzania

**George M. Ruhago** [1]*, **Frida N. Ngalesoni**[1], **Daudi Msasi**[2], **James T. Kengia**[3],
**Mathew Mganga**[3], **Ntuli A. Kapologwe**[3], **Majiliwa Mtoroki**[4], **Mavere A. Tukai**[5]

**1** Department of Development Studies, School of Public Health and Social Sciences, Muhimbili University of Health and Allied Sciences, Dar es Salaam, Tanzania, **2** Ministry of Health, Community Development, Gender, Elderly and Children, Dodoma, Tanzania, **3** President's Office Regional Administration and Local Government (PORALG), Dodoma, Tanzania, **4** Medical Stores Department, Dar es Salaam, Tanzania, **5** USAID Global Health Supply Chain Program Technical Assistance, Dar es Salaam, Tanzania

\* ruhagogm@gmail.com

**Data Availability Statement:** All data are reported within the paper.

**Funding:** The USAID- Global Health Supply Chain – Technical Assistance Tanzania provided financial

## Abstract

Tanzania's supply chain system is a complicated web of integrated and vertical systems, covering essential and vertical programs health commodities, laboratory and diagnostics, equipment, and supplies. Despite significant improvement in the supply chain over the decades, the availability of medicines has remained uneven. Therefore, identifying the cost of operating the supply chain is vital to facilitate allocation of adequate finances to run the supply chain. We adopted a three-step approach to costing, which included i) identification, ii) measurement, and (iii) valuation of the resource use. Two levels of the Tanzanian supply chain system were examined to determine the cost of running the supply chain by function. These included first the Medical Stores Department (MSD) central and zonal level, secondly the health service delivery level that include National, Zonal and regional hospitals and the Primary Health Care (District Hospital, health center and Dispensary). The review adopted the health system perspective, whereby all resources consumed in delivering health commodities were considered, resource use was then classified as financial and economic costs. The costing period was an average of two financial years, 2015/16 and 2016/17. The cost data were exchanged from Tanzania Shillings to 2017 US$ and then adjusted for inflation to 2020 US$. The study used the total sales reported in audited financial accounts for throughput value. The average annual costs of running the supply chain at the central MSD was estimated at USD$ 15.5 million and US$ 4.1 million at the four sampled MSD Zonal branches. There is a wide variation in annual running costs among MSD zonal branches; the supply chain's unit cost was highest in the Dodoma zone and lowest in the Mwanza zone at 26% and 8%, respectively. When examined on a cost-per-unit basis, supply chain systems operate at sub-optimal levels at health facilities at a unit cost of 37% per unit of commodity throughput value. There are inefficiencies in supply chain financing in Tanzania. Storage costs are higher than distribution costs, this may imply some efficiency gains. MSD should employ a "just in time" inventory model, reducing the inventory holding costs, including the high-expired commodities holding charge, this could be improved by increasing the order fill rate.

support of this work. The funders had no role in study design, data collection and analysis, decision to publish, or preparation of the manuscript.

**Competing interests:** The authors have declared that no competing interests exist.

# Background

Public health systems strive to improve the health outcome of the population. Central to this goal is a fully functioning health commodities supply chain system which is central to a public health system's performance and the care it provides to the public [1]. A reliable public sector supply chain ensures accessible, timely, affordable, and quality health commodities to the health care facilities. Efforts have been made to improve the health commodities supply chain in Tanzania. The Ministry of Health (MOH) has the mandate of stewardship, policy making and resource mobilization for the health commodity supply chain. Under devolution, the President's Office, Regional Administration and Local Government (PORALG) is responsible for coordinating and facilitating the local government authorities (LGAs), who are mandated to deliver primary health care services, including health commodities.

The Medical Stores Department (MSD) was established by the Act of Parliament no. 13 of 1993 [2]. MSD is a semi-autonomous department under the Ministry of Health (MoH) mandated with procurement, storage, and distribution of approved medicines and medical supplies required by public health services. MSD operates one central store, eight zonal stores covering the Tanzanian administrative zones—and two sales points within the large zones. Health facilities allocate funds in their budget and purchase health commodities from MSD. To improve performance of MSD and supply chain in Tanzania, MoH under the development partner support, established a national logistics management unit (LMU) and a national electronic logistics management information system (eLMIS), to assist in organizing, monitoring, and supporting all supply chain activities for all health commodities at MoH and MSD zones [3].

Several reviews of the Tanzania supply chain have been conducted in the last decade. Despite studies reporting significant gains in improving the performance of the public sector supply chain, the availability of essential health commodities has continued to remain uneven, with availability of essential medicines in the public sector health supply chain in Tanzania reported at about 56 percent, at the time of this study [4, 5]. One of the crucial factors of these frequent stock-outs is inadequate financing and operational inefficiencies [5–7]. However, the performance of the supply chain depends on the availability of adequate resources to finance health commodity procurement and cover the cost of running the supply chain at an acceptable performance standard. These costs, including those for procurement, clearing and forwarding, insurance, storage, distribution and staff salaries, etc., are estimated to be between 10% and 50% of the value of procured commodities [7]. Overlooking such costs may hamper the functioning of the supply chain, especially in resource constraint settings. Therefore, identifying the cost of operating the supply chain and the available financing options is vital to facilitate adequate resource allocation.

# Methods

## Ethics statement

Ethics approval for this study was obtained from the Institutional Review Board of Muhimbili University of Health and Allied Sciences. Verbal consent was obtained from all participants. All regulatory requirements to conduct this study in Tanzania were met.

## Study design and settings

This is a cross-sectional study employing quantitative approach to estimate the costs of running the supply chain in Tanzania from the public health system perspective. Costs were determined by key functions–procurement, storage, distribution, and management support—and by different levels of the public supply chain. Two levels of the Tanzanian supply chain system

were examined these included first the central MSD and Zonal MSD (Five zones i.e., Dar es salaam, Dodoma, Mbeya, Mtwara and Mwanza were sampled among the eight MSD zones to determine the average supply chain running cost at the Central and Zonal MSD. Second level was the health facilities, Tanzania has about four levels of health facilities the National Hospitals with one large National Hospital and five specialized hospitals i.e., Orthopedic and Trauma Hospital, Cardiac Hospital, Cancer Hospital, Mental Health Hospital, and the Infectious Disease Hospital (mainly specialized in Tuberculosis). The Zonal Hospitals composed of with four main zonal hospitals i.e., Bugando Medical Center in Mwanza for Lake Zone, KCMC for Northern Zone, Mbeya Zonal Hospital for Southern Zone, and Benjamin Mkapa Hospital for Central Zone. There are about 26 Regional hospitals and Primary Health Care facilities (Council Hospital, Health Center, and Dispensary).

## Sample selection

The multistage stratified non-probability sampling was employed. At the minimum the sample were set to include facilities MSD Central, MSD Zones and Health facilities at the National, Zonal, Regional and Primary Health Care level. The facilities were sampled based on level of services offered and geographical location i.e., urban/rural with a final list of 26 facilities. See Table 1 below for list of study facilities.

## The costing approach

The supply chain manager's handbook [8], complemented with detailed interviews with key personnel at the different levels, was used to identify all resources used. Measurement of identified resources followed activity-based costing method [9, 10]. This required examining the audited financial reports (post-expenditure report) and annual performance reports, asset registers, inventory records, and whenever necessary, physical counting of equipment and space measurement was done. The cost data were exchanged from Tanzania Shillings to 2017 US$ and then adjusted for inflation to 2020 US$, using Bank of Tanzania historical exchange rates [11] and the World Bank inflation rates [12]. Valuation of the measured resources employed opportunity cost approach.

**Table 1. Health facilities visited for supply chain costing.**

| Level | Number of facilities | Name of Facilities |
|---|---|---|
| | | **MSD** |
| **MSD Headquarter** | 1 | MSD Central |
| **MSD Zones** | 4 | **Dar es salaam, Dodoma, Mbeya, Mtwara and Mwanza** |
| | | **National, Zonal and Regional Hospitals** |
| **National** | 4 | Muhimbili National hospital, Muhimbili Orthopedic Institute (MOI), Mirembe hospital*, and Ocean Road Cancer Institute (ORCI) |
| **Zonal** | 2 | Bugando Medical Center (BMC), and Mbeya Referral hospital* |
| **Regional** | 2 | Temeke, Sekoutoure, and Kagera Regional Referral Hospitals (RRH) |
| | | **Primary Health Care** |
| **Council Hospital** | 5 | Mbagala District Hospital (DH), Nyamagana DH, Misungwi DH, Mbalizi District Designated hospital (DDH) and Izimbya DDH |
| **Health centers** | 5 | Kigamboni, Makongoro, Misasi, Ruanda and Inyala |
| Dispensar*ies* | 8 | Yombo Vituka, Kizuiani, Mahina, Ntulya, Isesye, Idunda and Mazinga |

* Due to incomplete data information from these facilities were excluded in the analysis.

**Perspective.** The study adopted the health system perspective, whereby all resources consumed in delivering health commodities were considered. The costing period was an average of two financial years, 2015/16 and 2016/17.

## Resource identification

All resources used were identified such as buildings, equipment, salaries, stationaries, travels and their associated daily subsistence allowances (DSA), meetings, building operations, fuel, and other vehicle maintenance expenses. The resource was identified based on the core functions of the supply chain; i) procurement, included, time spent on identifying reliable supplies of good quality health commodities, management of bids and tenders and execution of procurement contracts. ii) Storage included all resources utilized in receiving and warehousing of health commodities. iii) Distribution included costs incurred in the movement of health commodities from central store to zonal warehouses and zonal warehouses to health facilities, and office spaces used by the distribution staff and vehicle garages. iv) Management support included all costs incurred in supervising and coordination of all supply chain activities, v) Logistics Management Unit (LMU), all the costs incurred in conducting LMU activities at MSD zones were costed. vi) Serving patients, these included all costs incurred to dispense health commodities to the clients such as salaries for health workers dispensing the health commodities, building space, furniture etc.

## Resource measurement and valuation

Resource use was classified as financial and economic costs [13], the financial costs included all costs that are directly incurred by MSD or health facilities for health commodities delivery and the economic costs included all costs that are contribute in the delivery of health commodities but are not directly paid by MSD or health facilities such as donated goods and workers paid by development partners. Later the costs were grouped under capital or recurrent costs. Capital costs consist of buildings and equipment whose useful life exceeds a year, recurrent costs include salaries, stationaries, travels DSA, meetings, building operations, fuel, and other vehicle maintenance expenses. Capital costs were annuitized using a rate of 16%, which was the average interest rate for the two financial years (2014/15 and 2015/16) reported by the Bank of Tanzania [14]. We employed applicable life years based on World Health Organization (WHO) assumptions, for instance, 30 years for building, ten years for vehicles, five years for computers etc. [15]. Capital items costing less than US$50 (Tshs 100 000) were treated as recurrent costs.

To value the resources used in delivering the health commodities, we used several data sources, these included government salary structures for salaries, MSD price list for health commodities and medical supplies, Tanzania Government Procurement Services Agency tender prices for other non-medical equipment and supplies. The replacement cost for the building structure was approximated based on estimate from the Housing Finance in Africa Yearbook 7th edition [10]. All data were extracted using structured data collection guides and data extraction tools. The proportion of the number of staff at each of the cost centre as a percentage of total staff at each of the study site i.e. central and zonal MSD, as well as health facilities, was used as allocation key to apportion shared resources such as building space, supplies, building operations costs etc. [16].

## Data analysis

All cost data were entered, summarized, and analyzed using Excel. The total cost of delivering the health commodities was calculated for the different levels of the supply chain. The unit

costs of delivering the health commodities were computed as the total cost of delivering the health commodities over the supply chain throughput value i.e., the value of the health commodity delivered. The value of the items delivered was the total sales reported in MSD's audited financial accounts and the value of the stock of health commodities during the study period from the MSD statement of accounts. The total value of expired health commodities was divided to the total health commodities managed at each MSD level to determine the percentage of expired commodity to the total value of stock managed by MSD during the study period. During discussion with MSD management, they identified storage and management support as the cost centers incurring the costs of holding the expired commodities. The unit cost of running the supply chain at Central and Zonal MSD were calculated as the total cost of the supply chain at the zone level delivering the health commodity over the supply chain throughput value i.e., value of the health commodity delivered managed at each zone.

## Results

### Supply chain operation cost at central MSD

The average annual total costs of running the supply chain at central MSD in 2020 US$ was estimated at US$ 15.5 million. The through put value at MSD central was US$ 69.8 million. The average unit cost of operating the supply chain at central MSD was estimated at 22% per unit of commodity throughput value. Management support accounted for the highest percentage (39%) of the total costs at the central MSD, while storage (51%) and distribution (38.1%) were the highest costs driver at the zonal MSD (Table 2). The total value of health commodities managed at central MSD is USD$ 66.2 million.

### Supply chain operations costs at MSD zones

The Average total costs of running supply chain at the four sampled zones was US$ 4.1 Mil with an average cost of US$ 1.03 million per MSD Zone (Table 3). However, the costs varied considerably among the MSD zones, ranging from US$ 0.29 million in Mtwara to US$ 1.3 million in Dodoma Zone. The through put value at zonal MSD was US$ 30.6 million, Dar es salaam zone with the highest throughput value at US$ 9.2 million, and Mtwara zone with the lowest throughput value at US$ 3.1 million. The average unit cost of the supply chain at the zone level was at 13.4% per throughput value with the highest value recorded in the Dodoma zone (29%) and lowest in the Mwanza zone at 8% (Table 4).

### Supply chain cost at the health facility level

The average supply chain costs are larger, the higher the health care delivery level, due to lower throughput value at lower-level health facilities. The average unit costs per throughput was 12 percent. The value was higher at dispensary level (72%) compared to National/Zonal hospital level (8%) health care facilities (Table 5).

### Costs of holding expired commodities

There was the varying cost of holding expired commodities at MSD zones. These costs were significantly high in Dodoma zone at US$ 132,264 resulting in unit cost of per throughput value of 11% (Table 6).

## Discussion

The supply chain response to the health care needs depends on the availability of finances to procure as well as finances to run the supply chain the latter of which is often overlooked.

**Table 2. Average annual costs of running the supply chain at the central medical stores department (2020 US$).**

| Cost category | Procurement | | Storage | | Distribution | | Management Support | | Total Financial Cost | Total Economic Cost | Total cost | % |
|---|---|---|---|---|---|---|---|---|---|---|---|---|
| | Fin. | Eco. | Fin. | Eco. | Fin. | Eco. | Fin. | Eco. | | | | |
| **Capital** | | | | | | | | | | | | |
| **Buildings** | 0 | 0 | 1,752,365 | 798,909 | 14,281 | 0 | 69,478 | 0 | 1,836,124 | 798,909 | 2,635,033 | 17.0% |
| **Equipment** | 10,973 | 0 | 104,534 | 0 | 71,615 | 0 | 68,974 | 0 | 256,096 | 0 | 256,096 | 1.6% |
| **Vehicles** | 0 | 0 | 0 | 0 | 644,368 | 0 | 53,588 | 3,287 | 697,956 | 3,287 | 701,243 | 4.5% |
| **Training** | 12,219 | 1,281 | 16,582 | 1,738 | 11,346 | 1,189 | 48,002 | 5,031 | 88,149 | 9,238 | 97,387 | 0.6% |
| **Total capital costs** | **23,192** | **1,281** | **1,873,481** | **800,647** | **741,610** | **1,189** | **240,042** | **8,317** | **2,878,325** | **811,434** | **3,689,758** | **23.8%** |
| **Recurrent** | | | | | | | | | | | | |
| **Personnel** | 18,912 | 0 | 884,354 | 0 | 524,777 | 0 | 3,006,185 | 0 | 4,434,228 | 0 | 4,434,228 | 28.6% |
| **Supplies** | 51,014 | 0 | 66,695 | 0 | 82,934 | 0 | 113,833 | 0 | 314,477 | 0 | 314,477 | 2.0% |
| **Vehicle operations** | 0 | 0 | 0 | 0 | 475,679 | 0 | 233,543 | 0 | 709,223 | 0 | 709,223 | 4.6% |
| **Building operations** | 116,732 | 0 | 1,432,301 | 0 | 233,464 | 0 | 2,191,583 | 0 | 3,974,080 | 0 | 3,974,080 | 25.6% |
| **Meetings** | 0 | 0 | 0 | 0 | 0 | 0 | 241,015 | 0 | 241,015 | 0 | 241,015 | 1.6% |
| **Visits/Supervision** | 0 | 0 | 0 | 0 | 0 | 0 | 50,148 | 0 | 50,148 | 0 | 50,148 | 0.3% |
| **Port expenses** | 1,530,809 | 0 | 0 | 0 | 0 | 0 | 0 | 0 | 1,530,809 | 0 | 1,530,809 | 9.9% |
| **Inspection costs** | 238,417 | 0 | 0 | 0 | 0 | 0 | 0 | 0 | 238,417 | 0 | 238,417 | 1.5% |
| **Quality control/ assurance** | 85,239 | 0 | 0 | 0 | 0 | 0 | 0 | 0 | 85,239 | 0 | 85,239 | 0.5% |
| **Other** | 178,618 | 0 | 0 | 0 | 83,763 | 0 | 0 | 0 | 262,380 | 0 | 262,380 | 1.7% |
| **Total recurrent costs** | **2,219,741** | **0** | **2,383,349** | **0** | **1,400,618** | **0** | **5,836,308** | **0** | **11,840,017** | **0** | **11,840,017** | **76.2%** |
| **TOTAL COSTS** | **2,242,933** | **1,281** | **4,256,831** | **800,647** | **2,142,228** | **1,189** | **6,076,350** | **8,317** | **14,718,341** | **811,434** | **15,529,775** | **100%** |
| | **14%** | **0%** | **27%** | **5%** | **14%** | **0%** | **39%** | **0%** | **95%** | **5%** | **100%** | |
| **Total Sales and stock (US$)** | | | | | | | | | | | **69,798,585** | |
| **Unit Cost Per US$ Throughput** | | | | | | | | | | | **22.2%** | |

**Note:** Fin. = Financial costs and Eco. = Economic costs.

Therefore, estimating the costs of operating the supply chain is key in financing national public health commodity supply chain to ensure adequate and timely procurement, storage and delivery of health commodities to the last mile [17].

The results of this study indicate that the unit cost per commodity throughput value at central MSD was 22% and at zonal MSD was at 13%. Storage and distribution costs were shown to be the main contributors to the total costs. This is similar to what was observed in Ghana, where storage was the main cost driver in public run supply chain systems. The primary reasons for the high distribution costs could be the significant depreciated fleet resulting in additional distribution costs expended in fleet maintenance. One of the most discussed cost-cutting strategies is using Third-Party Logistics (3PLs, the strategy is likely to reduce supply chain costs and improving performance [18–20]. However, this option requires a thorough assessment of the need for (in whole or part) and the potential for distribution outsourcing. Government-run health commodity distribution systems, if anything, are expected to operate with social welfare in mind or, at worst, cost recovery [21]. Similarly, large and half empty warehouses provides an area for further efficiency improvement through increasing stock

**Table 3. The average annual costs of running the supply chain at four sampled zonal medical stores department by supply chain functions (2020 US$) (N = 4).**

| Cost category | Procurement | | Storage | | Distribution | | LMU | | Mgt Support | | Total Financial Cost | Total Economic Cost | Total cost | % |
|---|---|---|---|---|---|---|---|---|---|---|---|---|---|---|
| | Fin. | Eco. | Fin. | Eco. | Fin. | Eco. | Fin. | Eco. | Fin. | Eco. | | | | |
| **Capital** | | | | | | | | | | | | | | |
| Buildings | 0 | 0 | 383,847 | 1,151,034 | 112,967 | 280,815 | 12,530 | 41,050 | 52,301 | 0 | 561,643 | 1,472,899 | 2,034,543 | 49.4% |
| Equipment | 0 | 0 | 21,665 | 0 | 21,272 | 0 | 0 | 1,944 | 4,263 | 0 | 47,199 | 1,944 | 49,143 | 1.2% |
| Vehicles | 0 | 0 | 0 | 0 | 317,805 | 17,233 | 0 | 4,035 | 13,401 | 0 | 331,206 | 21,269 | 352,474 | 8.6% |
| Training (non-recurrent) | 820 | 0 | 1,112 | 0 | 761 | 0 | 0 | - | 3,220 | 0 | 5,913 | 0 | 5,913 | 0.1% |
| **Total capital costs** | 820 | 0 | 406,624 | 1,151,034 | 452,804 | 298,049 | 12,530 | 47,029 | 73,183 | 0 | 945,961 | 1,496,112 | 2,442,073 | 59.3% |
| **Recurrent** | | | | | | | | | | | | | | |
| Personnel | 1,991 | 0 | 117,095 | 0 | 239,861 | 0 | 8,429 | 20,235 | 44,040 | 0 | 411,417 | 20,235 | 431,652 | 10.5% |
| Supplies | - | 0 | 50,556 | 0 | 179,741 | 0 | - | 2,546 | 21,471 | 0 | 251,769 | 2,546 | 254,315 | 6.2% |
| Vehicle operations | - | 0 | - | 0 | 524,941 | 0 | - | 7,735 | 14,463 | 0 | 539,404 | 7,735 | 547,140 | 13.3% |
| Building operations | - | 0 | 229,187 | 0 | 9,801 | 0 | 9,788 | 51 | 93,634 | 0 | 342,411 | 51 | 342,462 | 8.3% |
| Meetings | - | 0 | - | 0 | - | 0 | 0 | - | 50,441 | 0 | 50,441 | 0 | 50,441 | 1.2% |
| Visits | 947 | 0 | - | 0 | - | 0 | 0 | - | | 0 | 947 | 0 | 947 | 0.0% |
| Quality assuarance | 7,109 | 0 | - | 0 | 14,154 | 0 | 0 | 1,349 | 25,142 | 0 | 46,406 | 1,349 | 47,755 | 1.2% |
| Other | 10,048 | - | 396,838 | - | 968,499 | - | 18,217 | 31,916 | 249,193 | - | 1,642,796 | 31,916 | 1,674,711 | 40.7% |
| **Total recurrent costs** | 10,867 | 0 | 803,462 | 1,151,034 | 1,421,304 | 298,049 | 30,747 | 78,945 | 322,377 | 0 | 2,588,756 | 1,528,028 | 4,116,784 | |
| % | 0.3% | 0% | 19.5% | 28.0% | 34.5% | 7.2% | 0.7% | 1.9% | 7.8% | 0% | 62.9% | 37.1% | | 100.00% |

**Note:** Fin. = Financial costs and Eco. = Economic costs.

levels. During the period of this study the national health commodity stock availability was about 56 percent [5].

We observed wide variation in annual running costs among MSD zones. Since storage was one of the key cost drivers it was assumed that zones recording high cost, can be attributed to running Warehouses in the Box (WIB). WIB are ready-made metal warehouse that can be assembled within a short period of time with shipping and installation taking about six to nine months. Normally the WIB are insulated, however we hypothesised that, running costs was higher in Dodoma due to using WIB, Dodoma is a relatively hot region but when compare to Mbeya in the southern highlands and hence relative cooler weather, there was no any significant difference in the warehouse running cost. It therefore, remains an empirical question into exploring other factors contributing to the efficient running of supply chain operations in one zone compared to the other.

**Table 4. Average unit costs of supply chain operations at the MSD zones (2020 US$).**

| Cost category | Mwanza | Mtwara | Mbeya | Dodoma | Dar es Salaam | Total |
|---|---|---|---|---|---|---|
| Total SC Operating Costs (US$) | 752,512 | 293,505 | 921,216 | 1,269,718 | 879,833 | 4,116,784 |
| Total Sales and stock value(US$) | 9,154,580 | 3,116,492 | 4,709,505 | 4,424,747 | 9,231,453 | 30,636,777 |
| Unit Cost Per US$ Throughput | 8% | 9% | 20% | 29% | 10% | 13.4% |

**Table 5. Average annual supply chain operation costs, by health care facilities level (2020 US$) (N = 26).**

| Function | National/Zonal | Regional | Council | Health Centers | Dispensaries |
|---|---|---|---|---|---|
| Procurement | 92,447 | 8,529 | 5,626 | 4,242 | 2,230 |
| Storage | 73,929 | 51,852 | 14,911 | 8,386 | 5,755 |
| Serving clients | 64,458 | 43,437 | 17,378 | 7,540 | 4,861 |
| Management Support | 52,833 | 33,799 | 20,396 | 5,199 | 3,986 |
| **Total SC costs (US$)** | **283,668** | **137,617** | **58,311** | **25,367** | **16,833** |
| Commodity value (US$) | 3,578,766 | 468,058 | 165,038 | 60,147 | 23,429 |
| Unit cost per US$ throughput | 8% | 29% | 35% | 42% | 72% |

The low throughput value in the Dodoma zone indicates the possibility that the zone is operating at inadequate levels, notwithstanding its function as an MSD central buffer. Therefore, assuming that funding will stabilize soon, there is a need for a thorough capacity utilization analysis focusing on cubic utilization. The results of such an analysis may imply rethinking how zones serve their clients from concentrating on regional to serving clients by districts.

The findings of this study imply that the supply chain system might be operating at suboptimal levels at lower lever facilities i.e., the primary health care, when examined based on average supply chain costs per unit of health commodities managed, which was found to be 37% of the value of health commodities managed at this level. Apart from this high unit costs estimated, issues of poor quality of data pertaining value of health commodities procured/delivered at this level cannot be underscored as such these results should be interpreted with caution. For instance, it was difficult to obtain data on the value of vertical program commodities i.e., the health commodities for programs that are operated at the national level, these includes HIV/AIDS, Tuberculosis, Malaria and Family Planning commodities among the few. Unit costs are likely to be lower than what is portrayed in this study due to the management of vertical program commodities.

Assuming the cost estimates in the 'as is" scenario were adequate, the average annual order fill rates from the MSD zones to health facilities, for health commodities requested by health facilities at the time of this study was about 56% [5]. Increasing the quantities of commodities managed at the health facilities to meet the demand would significantly reduce these costs. These endeavors would be facilitated by the new GoT initiatives of implementing the Direct Facility financing (DHFF) [22, 23], which empowers health facilities to manage their funds and thus procure health commodities as per their needs.

The direct costs associated with inventory are the monetary value of safety stock required to avoid stock-outs and the value of products that expire. The costs of holding the latter are usually ignored. In this study, it was found that there were very high inventory holding costs at some of the MSD zonal branches. There may be several reasons for expired commodities in a supply chain ranging from uncoordinated donations, poor quantification to inadequate

**Table 6. Average costs of holding expired commodities at the medical stores department, (2020 US$).**

| Total Costs (USD$ Million) | Mwanza | Mtwara | Mbeya | Dodoma | Dar es Salaam |
|---|---|---|---|---|---|
| **Total Supply Chain Costs** | 752,512 | 293,505 | 921,216 | 1,269,718 | 879,833 |
| **Total Costs of Holding Expired Commodities (TEC)** | 5,872 | 3,973 | 59,759 | 139,525 | 29,714 |
| **Unit Costs** | | | | | |
| **TEC as a % of the Total Supply Chain Costs** | 1% | 1% | 6% | 11% | 3% |

inventory control management. The literature indicate that the value of expired commodities is often a small percentage of the total value of items [24], however this resources could have been used efficiently to strengthen other areas of the health commodity supply chain. Further, the costs of disposing of these drugs can be enormous. Often, Government regulations fail to recognize the urgency in disposing of these commodities, leading them to accumulate in the storage areas for years.

Excessive expired commodities are not new; as such, recommendations from several reports have led to the development of donation guidelines [25] to tackle the situation. Engaging all stakeholders such as the Tanzania Medicine and Medical Devices Authority (TMDA) to ensure such regulations are implemented if these vast costs are to be curtailed. The MoH, in collaboration with the Ministry of Finance (MoF), could review the rules on disposal of health commodities to enable timely clearance of expired items. Again, implementing the "just in time" inventory model would also curtail stocking of large amount of inventory of slow-moving inventory, which ultimately leads to expired health commodities. A previous study in similar settings has shown that, engagement of all the actors in the health system is crucial in supply chain system redesign [26].

## Limitations

The accuracy and quality of these cost data remains questionable in a low income setting like Tanzania hence prompting interpretation of these results with caution [27]. The quality of some of the information cannot be affirmed with certainty; however, efforts were made to use data from multiple sources and from databases of reputable organizations such as WHO and official GoT documents.

## Conclusions

The findings of this study have highlighted that there is wide variation in public supply chain running costs among the zonal branches of the Medical Stores Department (MSD) and health facilities by level of service. Storage and distribution are the highest cost for the health commodities supply chain and should be a target for efficiency improvement. The price of holding expired commodities was significant, employing interventions such as the "just in time" inventory model would curtail the stocking of large amount of slow-moving inventory, which ultimately hence lead to expired health commodities. Supply chain systems are operating at sub-optimal levels. This could mainly be due low order fill rate. Increasing the quantities of items managed at the health facilities to meet the demand would significantly improve supply chain efficiency at this of service.

## Acknowledgments

MoHCDGEC and PORALG for providing administrative support during data collection and allowing the authors to use the data in this study.

## Author Contributions

**Conceptualization:** George M. Ruhago, Frida N. Ngalesoni, Daudi Msasi, James T. Kengia, Mathew Mganga, Ntuli A. Kapologwe, Majiliwa Mtoroki, Mavere A. Tukai.

**Data curation:** George M. Ruhago, Frida N. Ngalesoni.

**Formal analysis:** George M. Ruhago, Frida N. Ngalesoni, Mavere A. Tukai.

**Methodology:** George M. Ruhago, Frida N. Ngalesoni, James T. Kengia, Mathew Mganga, Ntuli A. Kapologwe, Majiliwa Mtoroki, Mavere A. Tukai.

**Writing – original draft:** George M. Ruhago.

**Writing – review & editing:** George M. Ruhago, Frida N. Ngalesoni, Daudi Msasi, James T. Kengia, Mathew Mganga, Ntuli A. Kapologwe, Majiliwa Mtoroki, Mavere A. Tukai.

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
