## [Decision Letter · Decision Letter 0]

21 Jun 2022

PGPH-D-22-00641

Estimating the supply chain costs and the existing financing mechanisms in Tanzania

Dear Dr. Ruhago,

Thank you for submitting your manuscript to PLOS Global Public Health. After careful consideration, we feel that it has merit but does not fully meet PLOS Global Public Health’s publication criteria as it currently stands. Therefore, we invite you to submit a revised version of the manuscript that addresses the points raised during the review process.

Please submit your revised manuscript by . If you will need more time than this to complete your revisions, please reply to this message or contact the journal office at globalpubhealth@plos.org. Please include the following items when submitting your revised manuscript:

We look forward to receiving your revised manuscript.

Kind regards,

Hassan Haghparast Bidgoli

Academic Editor

Journal Requirements:

a) State the initials, alongside each funding source, of each author to receive each grant.

2. Please update your Competing Interests statement. If you have no competing interests to declare, please state: “The authors have declared that no competing interests exist.”

3. In the online submission form, you indicated that “The data will be available on request from the authors.”. All PLOS journals now require all data underlying the findings described in their manuscript to be freely available to other researchers, either 1. In a public repository, 2. Within the manuscript itself, or 3. Uploaded as supplementary information.

Additional Editor Comments (if provided):

- Please provide more explanation on sampling methods and sample size calculation

- Please explain clearly what data are used in the costing

- Please improve policy implications of the findings based on the reviewers' comments

Reviewers' comments:

Reviewer's Responses to Questions

**Comments to the Author**

1. Does this manuscript meet PLOS Global Public Health’s publication criteria? Is the manuscript technically sound, and do the data support the conclusions? The manuscript must describe methodologically and ethically rigorous research with conclusions that are appropriately drawn based on the data presented.

Reviewer #1: Yes

Reviewer #2: Yes

2. Has the statistical analysis been performed appropriately and rigorously?

Reviewer #1: N/A

Reviewer #2: N/A

3. Have the authors made all data underlying the findings in their manuscript fully available (please refer to the Data Availability Statement at the start of the manuscript PDF file)?

Reviewer #1: Yes

Reviewer #2: Yes

4. Is the manuscript presented in an intelligible fashion and written in standard English?

Reviewer #1: Yes

Reviewer #2: Yes

5. Review Comments to the Author

Reviewer #1: GENERAL COMMENTS:

This paper describes costs for the public health care supply chain in Tanzania. The findings highlight significant variation in costs between different regions and zonal branches of the Medical Stores Department (MSD). Storage – as opposed to transport - is a significant cost.

Generally, the paper provides a valuable contribution to the literature on supply chain costs and components thereof. The methods used for costing are clearly described and appropriate. The text is generally easy to follow.

However, I would have liked to see more clarity from the authors in the Discussion and specifically in terms of policy implications. The authors argue that “identifying the cost of operating the supply chain and the available financing options is vital to facilitate adequate resource allocation for the supply chain.”(rows 80-81). The question then needs to be: do the costs presented here reflect a well-functioning supply chain, or not? How will this study help decision makers and planners?

Overall the Discussion section would benefit from being more clearly structured. Right now it has many threads, and it is not very clear to the reader what are the main points that the authors which to convey.

While the authors claim that the cost per unit would drop if more commodities were to flow through the system, clearly the total cost will increase. If there are budget constraints, how would the authors suggest these be approached? The authors attempt to make a few suggestions on how the current budget can be used more efficiently - such as just-in-time, 3PL, and resource allocation formula. It would be helpful if these suggestions could be grouped and presented more clearly to the reader in relation to the different cost elements. When the authors suggest developing a resource allocation formula (row 270), what exactly would they propose? Can they point to examples in the literature?

In terms of the proposal for a “just in time” model, it would be interesting to see the authors reflecting on the covid19 experience and whether this had any implications for Tanzania’s supply chain, and specifically whether a “just-in-time” model would be a less optimal solution during a global crisis when global supply chains are disrupted.

Finally, the last two paragraphs in the Discussion would actually fit better in the introduction. It would also be helpful to add information upfront in the paper on what % share of the health system is public. (row 287: “the majority of the Tanzanian population” – can actual numbers be added?)

SPECIFIC COMMENTS ON METHODS, RESULTS AND TABLES:

Are costs presented here economic or financial costs? The methods mention both.

Table 1 should be expanded to include a column showing the number of facilities per level.

Table 4: how did you define the cost of serving clients? Please describe more clearly what this refers to.

Coherence of tables 3 and 4. It is unclear how the column headings in table 3 map to the rows/functions in table 4. Can the authors explain this in the paper?

Generally, the building costs look really high. Is this because of a 16% discount rate? Are the buildings newly constructed? This high cost should be explained and discussed in the paper.

SPECIFIC COMMENTS ON EDITING:

Title of the manuscript: The title of the paper could be made more specific to inform readers: suggest to add the words “health sector” and “public/government” to the title.

References: Some references are incomplete as they do not contain information on the publishers. – for example reference (8) is a USAID document and this should be added to the reference information.

Moreover reference (8) does not seem to be quoted properly as the text cites a commodity cost mark-up of 10-50% of value, and not 50% as written here by the authors (row 77 in the draft manuscript).

(see https://www.ghsupplychain.org/sites/default/files/2019-07/financing_service_fees.cfm_.pdf )

Here I would therefore suggest that the authors change their text to read 10-50% or reword as “…up to 50%).

I have a similar comment on editing for reference 13 which is another USAID document and which has a recommended citation: “McCord, Joseph, Marie Tien, and David Sarley. 2013. Guide to Public Health Supply Chain Costing: A Basic Methodology. Arlington, Va.: USAID | DELIVER PROJECT, Task Order 4.” It is unclear why the authors have not used the recommended citation.

Reviewer #2: Abstract

The methods section should provide brief information on the data sources for the analysis. It should also mention the levels of the health system the costing was done. And also please mention the types of costs presented (financial or economic costs).

There is a typo in line 45 which says two financials was. There are several other typos in the manuscript which should be corrected.

It is not clear how the determination that the operations are sub-optimal or inefficient as alluded to in line 49 and 51 was made.

The conclusion introduces new findings of information not provided in the results. New findings should not be presented in the conclusion.

Main manuscript

Methods

Line 98: It is not clear exactly what stratified purposive convenient sampling is.

Table 1 is not very helpful as readers who don’t know the Tanzania health system wont know any of those names or their significance or location. Why where those facilities selected? What supply chain are conducted at each of those levels?

How was the sample size determined?

The authors mentioned they used data from budgets and financial expenditures. Which data were from budgets and which ones were from financial reports? When both data existed, how were differences reconciled?

The methods section is generally weak and it is not clear what exactly was costed and where the data for each were obtained from or the methods for the data analysis.

How were costs for shared resources allocated?

Results

What is included in procurement costs – are commodity costs included or excluded? In the results tables, distribution costs include buildings. Please explain what building costs are included in the distribution function? Again these things should have been clarified in the methods.

The tables should include sample sizes and also be clear that these are economic costs (if they are).

The logistics management unit costs are presented for zonal medical stores but not for the central medical store. Why is this so?

Table 4 shows total costs of sales. What is being sold by the medical stores?

Line 183 mentions costs of holding expired commodities. The methods don’t explain how this was costed. Please add this detail to the methods.

Discussion

The discussion section is weak. There are statements where the reader does not see the linkage to the results and some of them seem more like views of the authors rather than discussion backed by the findings presented. For example, line 196 to 198 seem political and targeted to some stakeholder and hence not objective. Another example of non-objective statements is line 226 – it is not obvious that 3PL are cost-cutting, there are studies that have shown them to be more expensive and that is part of the reason they are not implemented.

It is not clear what running warehouses in the box means as alluded to on lines 206 to 209.

Line 254 to 260 should be backed by benchmarks which validate the statement. There are very few citations in the discussion, and this is a weakness as statements should be backed by facts from other studies.

The discussion section should include a section with the limitations of the study.

6. PLOS authors have the option to publish the peer review history of their article (what does this mean?). If published, this will include your full peer review and any attached files.

**Do you want your identity to be public for this peer review?** For information about this choice, including consent withdrawal, please see our Privacy Policy.

Reviewer #1: No

Reviewer #2: No

---

## [Decision Letter · Decision Letter 1]

29 Sep 2022

PGPH-D-22-00641R1

The public Sector Supply Chain Costs in Tanzania

Dear Dr. Ruhago,

Thank you for submitting your manuscript to PLOS Global Public Health. After careful consideration, we feel that it has merit but does not fully meet PLOS Global Public Health’s publication criteria as it currently stands. Therefore, we invite you to submit a revised version of the manuscript that addresses the points raised during the review process.

We look forward to receiving your revised manuscript.

Kind regards,

Hassan Haghparast Bidgoli

Academic Editor

Journal Requirements:

3. Please ensure that the Title in your manuscript file and the Title provided in your online submission form are the same.

Additional Editor Comments (if provided):

Reviewers' comments:

Reviewer's Responses to Questions

**Comments to the Author**

1. If the authors have adequately addressed your comments raised in a previous round of review and you feel that this manuscript is now acceptable for publication, you may indicate that here to bypass the “Comments to the Author” section, enter your conflict of interest statement in the “Confidential to Editor” section, and submit your "Accept" recommendation.

Reviewer #1: (No Response)

Reviewer #2: All comments have been addressed

2. Does this manuscript meet PLOS Global Public Health’s publication criteria? Is the manuscript technically sound, and do the data support the conclusions? The manuscript must describe methodologically and ethically rigorous research with conclusions that are appropriately drawn based on the data presented.

Reviewer #1: Yes

Reviewer #2: Yes

3. Has the statistical analysis been performed appropriately and rigorously?

Reviewer #1: N/A

Reviewer #2: Yes

4. Have the authors made all data underlying the findings in their manuscript fully available (please refer to the Data Availability Statement at the start of the manuscript PDF file)?

Reviewer #1: Yes

Reviewer #2: (No Response)

5. Is the manuscript presented in an intelligible fashion and written in standard English?

Reviewer #1: Yes

Reviewer #2: (No Response)

6. Review Comments to the Author

Reviewer #1: Thank you for allowing me to review the revised version of this manuscript.

Detailed accounts of costs associated with health commodity supply chains are rare in the published literature and this paper provides a valuable contribution.

The authors have addressed most of the reviewer comments and this second version of the manuscript is much improved. I recommend that a few more changes be made to improve the manuscript overall and increase its usefulness and readability. Most of the changes refer to clarity around methodology and presenting results more coherently such that the reader can grasp the full picture.

Specific comments:

1. Background section page 3 rows 85-87. There is a claim that despite studies reporting significant gains in improving the performance of the public sector supply chain, the availability of essential health commodities has continued to remain uneven. Could references be added to sustain this claim?

2. Methods: Some lack of clarity on the methods on page 6: row 151 where the authors mention having used budget-based estimates, but the following sentence indicates that expenditure reports were used. Here there is lack of clarity. Did the study look at budgets (pre-expenditure), expenditure data (post-expenditure), or both? Additional clarification is needed here with more specific detail (expenditure reports at the level of facility, district? Etc)

3. Methods: page 7, rows 176-178: it is unclear what costs are included under category (vi) serving patients. Can the authors give concrete examples here.

4. Methods: under resource measurement and valuation, page 7 row 180: “Resource use was classified as financial and economic cost”. However it appears that authors have only used one classification and valuation approach, and not a dual approach. In a dual approach financial and economic costs would be valued and reported separately, and their amounts would differ. Generally, financial costs are actual monetary expenditures for resources, such as staff salaries, facility rent, and medical supplies. The economic costs, on the other hand, are valued differently, according to their opportunity costs including those resources that are not paid for (but must be monetized), such as volunteer time, caregiver time, and donated space and equipment. Are the authors certain that the costs shown in the paper represent both economic and financial costs? Or is it rather just financial costs being presented here?

5. Results: there is an issue with the overall numbering of tables: there are two tables labelled as Table 4.

6. Some additional work is needed by the authors to explain how the different tables link to each other. If I understand correctly, there is no overall results table for costs at all levels? Reading the first sentence in the Results section, I assume that the total costs for the public health sector supply chain (please use the same terminology as in the title paper here) is = 15.5 + 4.1 = 19.6 million? Is there a way that you could combine tables 2 and 3 and show this combined cost somewhere in a table?

7. Results: page 9 row 224: the value reported here (23%) does not appear in the table referred to. Suggest that the 23% value be added to Table 2. Or should it rather be added to table 4/5?

8. Results: table 4 (page 10): I would suggest to add one more column with Total values for the country (national /average values). This will make the paper more helpful for other researchers.

9. Results: table 4 (page 11): I would suggest to add one more column with Total values for the country (national/ average values). This will make the paper more helpful for other researchers. I would assume the value would then be 23% in the bottom row (? See comment 7 above). However I am not clear on what data the tables actually present.

10. Results: table 5 (page 11): I would suggest to add one more column with Total values for the country (national/ average values). This will make the paper more helpful for other researchers.

11. Results: I would strongly suggest to remove the words “economic and financial (costs)” from all the table headings. The approach should be clear from the methods section.

12. Discussion section, page 12, row 275: I would urge the authors to avoid sweeping statements like “For obvious reasons…” as the reasons are not obvious to the reader.

13. Discussion section, page 13, rows 307-309: “it may be that…… empty warehouses…..”. Can the authors please include evidence or references to evidence that warehouses are empty.

14. Discussion section, page 13, rows 313: “The findings of this study imply that the supply chain might be operating at suboptimal levels” – why? Could you please explain what are the specific findings that point to this? Is it linked to costs?

15. Discussion section, page 13, rows 319: the term “vertical program” requires more explanation as readers may not know what this means.

16. Discussion section, page 13, rows 320: the term “annual order fill” requires more explanation as readers may not know what this means.

17. Conclusion section: The authors argue that the supply chain should be a target for cost reduction. I do not see how this conclusion was made based on data presented in the paper, as a 23% overhead is within the 10-50% range presented in the background section. Perhaps cost reduction is the wrong term, as it may lead to associations around budget cuts; and “efficiency gains” might be more appropriate terminology.

18. Overall, it seems that with the revision process some additional editing may be needed to check language and grammar carefully before making the final submission.

Reviewer #2: The authors have adequately addressed my comments.

7. PLOS authors have the option to publish the peer review history of their article (what does this mean?). If published, this will include your full peer review and any attached files.

**Do you want your identity to be public for this peer review?** For information about this choice, including consent withdrawal, please see our Privacy Policy.

Reviewer #1: No

Reviewer #2: No

---

## [Editor Report · Decision Letter 2]

28 Oct 2022

The public health sector supply chain costs in Tanzania

PGPH-D-22-00641R2

Dear Dr Ruhago,

We are pleased to inform you that your manuscript 'The public health sector supply chain costs in Tanzania' has been provisionally accepted for publication in PLOS Global Public Health.

Best regards,

Hassan Haghparast Bidgoli

Academic Editor
